# Soil microbiome analysis reveals effects of periodic waterlogging stress on sugarcane growth

Onnicha Leelastwattanagul[1], Sawannee Sutheeworapong[1], Ahmad Nuruddin Khoiri[2], Sudarat Dulsawat[1], Songsak Wattanachaisaereekul[1,3], Anuwat Tachaleat[1], Thanawat Duangfoo[1], Prasobsook Paenkaew[1], Peerada Prommeenate[4], Supapon Cheevadhanarak[1,2]*, Jiraporn Jirakkakul[1]*

1 Pilot Plant Development and Training Institute, King Mongkut's University of Technology Thonburi, Bangkok, Thailand, 2 Bioinformatics and Systems Biology Program, School of Bioresources and Technology and School of Information Technology, King Mongkut's University of Technology Thonburi, Bangkok, Thailand, 3 School of Food Industry, King Mongkut's Institute of Technology Ladkrabang, Bangkok, Thailand, 4 Biochemical Engineering and Systems Biology Research Group, National Center for Genetic Engineering and Biotechnology, National Science and Technology Development Agency at King Mongkut's University of Technology Thonburi, Bangkok, Thailand

☯ These authors contributed equally to this work.
* supapon.che@kmutt.ac.th (SC); jiraporn.jir@kmutt.ac.th (JJ)

**Data Availability Statement:** 16S rRNA and ITS gene sequence data associated with this article have been deposited at NCBI under BioProject accession number: PRJNA658446 with SRA

## Abstract

Sugarcane is one of the major agricultural crops with high economic importance in Thailand. Periodic waterlogging has a long-term negative effect on sugarcane development, soil properties, and microbial diversity, impacting overall sugarcane production. Yet, the microbial structure in periodically waterlogged sugarcane fields across soil compartments and growth stages in Thailand has not been documented. This study investigated soil and rhizosphere microbial communities in a periodic waterlogged field in comparison with a normal field in a sugarcane plantation in Ratchaburi, Thailand, using 16S rRNA and ITS amplicon sequencing. Alpha diversity analysis revealed comparable values in periodic waterlogged and normal fields across all growth stages, while beta diversity analysis highlighted distinct microbial community profiles in both fields throughout the growth stages. In the periodic waterlogged field, the relative abundance of Chloroflexi, Actinobacteria, and Basidiomycota increased, while Acidobacteria and Ascomycota decreased. Beneficial microbes such as *Arthrobacter*, *Azoarcus*, *Bacillus*, *Paenibacillus*, *Pseudomonas*, and *Streptomyces* thrived in the normal field, potentially serving as biomarkers for favorable soil conditions. Conversely, phytopathogens and growth-inhibiting bacteria were prevalent in the periodic waterlogged field, indicating unfavorable conditions. The co-occurrence network in rhizosphere of the normal field had the highest complexity, implying increased sharing of resources among microorganisms and enhanced soil biological fertility. Altogether, this study demonstrated that the periodic waterlogged field had a long-term negative effect on the soil microbial community which is a key determining factor of sugarcane growth.

accession numbers: SRR14677846-
SRR14679243.

**Funding:** This research project is supported by
Thailand Science Research and Innovation (TSRI)
Basic Research Fund: Fiscal year 2023 under
project number: 181932. The funder did not
participate in the study's design, data collection,
analysis, the decision to publish, or the
manuscript's preparation.

**Competing interests:** The authors have declared
that no competing interests exist.

## Introduction

Sugarcane (*Saccharum officinarum* L.) is one of the major agricultural crops in Thailand with high economic importance. Sugarcane is cultivated in 47 provinces and covers about 8% of the total agricultural land of Thailand [1], however, some areas are in lowlands and face the periodic waterlogging problem regularly. Waterlogging is a situation of land that is temporarily or permanently saturated with water. Temporary waterlogging not only strongly impacts plant growth, development, and yield, but also causes the shift of microbial profile of agricultural land [2,3]. This effect leads to a loss of production and productivity in sugarcane [4–6]. Sugarcane has four growth phases: germination and emergence (1 month after planting), tillering and canopy establishment (at 1.5–4 months), grand growth (at 4–9 months), and maturation or ripening (at 9–12 months) [7,8]. During the rainy season in Thailand (May–September), periodic waterlogged could occur in some agricultural fields. Thus, this unfavorable condition affects the growth and yield of sugarcane due to a reduction in photosynthesis, root development, leaf area, tiller production, stalk height, and sucrose yield [9–11].

The adverse effect of waterlogging can be reduced by drainage systems, soil nutrient addition, and plant breeding for waterlogging-tolerant traits [2]. However, the installation of drainage systems and fertilizer application are high-cost investments when compared to microbiome management for sustainable soil fertility improvement [12]. Microbiome management should be considered a sustainable approach for reducing the impact of waterlogging. Soil is a complex environment colonized by a wide range of microorganisms, including archaea, bacteria, and fungi that interact with each other. They play a crucial role in maintaining soil health and fertility. Soil microbes are involved in key ecosystem services such as decomposition, mineralization, and aggregate formation. These microbial communities have been associated with the overall soil quality and agricultural productivity [13].

The rhizosphere, which is defined as the narrow zone of soil around the roots of living plants, is known for a higher microbial biomass and activity influenced by plant secondary metabolites compared to the surrounding bulk soil [14]. This area was reported to be colonized by plant growth-promoting rhizobacteria (PGPR). PGPR have the capability to produce plant regulators such as auxin and cytokinin which promote plant growth and productivity [15]. In addition, they can also protect crops from soil-borne pathogens and other stresses [13,16,17]. A complex interplay between the host plant and environmental conditions, e.g., growth stage, soil compartment, and stresses, caused significant changes in microbial diversity and their functions [18,19]. Specifically, waterlogging stress can lead to significant shifts in the composition of the rhizosphere microbiota, affecting both beneficial and detrimental microbial taxa. For instance, waterlogging stress can increase the proportion of bacteria with anaerobic respiratory capabilities within certain phyla like Firmicutes and Desulfobacterota, which might include plant-detrimental taxa such as Clostridium or Geobacter. On the contrary, waterlogging stress can decrease the proportion of Actinobacteria and Proteobacteria, particularly in families Rhizobiaceae and Xanthobacteraceae, which are considered beneficial for plants. This alteration in the microbial community composition can have repercussions on the health and resilience of plants [20]. Waterlogging can reduce the growth and development of sugarcane, leading to a 15–45% reduction in sugarcane yield [21]. The response of sugarcane plants to waterlogging stress has been reported in terms of growth, development, yield, and quality [5,22,23]. However, the soil microbial diversity related to sugarcane growth in agricultural fields is still poorly studied. Thus, the focus here was on the effect of periodic waterlogging stress on the growth of sugarcane and the structure of microbial communities at various sugarcane growth stages and soil compartments. The comparative study of microbiome profiles between normal and periodic waterlogging in sugarcane fields could be a promising approach

to reveal the impact of periodic waterlogging on soil bacterial and fungal diversity. The responsible taxa could then be identified as biomarkers for application in improving soil quality in waterlogged land to reduce the effect of waterlogging on sugarcane growth in the fields.

## Materials and methods

### The description of sugarcane field sites

The distance between the normal and periodic waterlogged fields was approximately 286 m, with an aisle of approximately 500 m separating them. The geographical coordinates for the normal and periodic waterlogged fields were recorded as follows: 13.7452153, 99.907871 for the normal field and 13.7446511, 99.9050294 for the periodic waterlogged field, located in Ratchaburi, Thailand. The soil type of the normal field was clay loam, while the periodic waterlogged field was clay soil. Both the normal and periodic waterlogged fields were managed in the same way, with the application of chemical fertilizers (N-P-K with a formula of 16-8-8) as a base dressing, followed by the application of the 20-8-20 formula after planting for 5 months. Both areas were equipped with a drip irrigation system. Initially, the sugarcane was watered once a week after planting and continued to receive watering until reaching two months of age. Subsequently, water was provided every 10 days until the sugarcane reached four months of age, after which it was watered once a month. However, irrigation would be suspended in the event of rainfall.

### Soil sample collection

Samples of rhizosphere (RHI), bulk soil around the field (BAF), and bulk soil around the tree (BAT) were collected for 5, 3, and 3 replicates, respectively, at each of the growth phases, i.e., tillering (TP; 3 months), grand growth (GP; 7 months), and ripening (RP; 9 months), in both normal and periodically waterlogged sugarcane fields (Table 1). BAF samples were collected from a 1-meter radius around the sugarcane field at a depth of 15 cm from the topsoil layer. BAT samples were collected from four points around the sugarcane stool, approximately 30–50 cm away from the stool and at a depth of 15 cm from the topsoil layer. RHI samples were obtained by collecting the soil tightly attached to sugarcane roots that were dug approximately 20 cm below the soil surface. All collected soil samples were chilled on ice before being transported to the laboratory. Soil samples were preprocessed and stored for further use according to Bulgarelli *et al.* (2012) [24]. Briefly, each RHI sample was suspended in a 0.85% (w/v) NaCl buffer in a 50 ml falcon tube and shaken for 10 minutes at room temperature, followed by root removal. The remaining RHI fraction was precipitated by centrifugation at 5,000 rpm for 15 minutes and the supernatant was discarded. The samples were then stored at −80˚C until DNA extraction.

### Soil chemical property analysis

Surface soil samples (0–20 cm) were collected from fifteen randomly distributed positions at the RP stage in both normal and periodic waterlogged fields. For each condition, soil samples

**Table 1. The number of samples collected from each sampling site for this study.**

| Stage | TP | | | GP | | | RP | | |
|---|---|---|---|---|---|---|---|---|---|
| Plantation Field | BAF | BAT | RHI | BAF | BAT | RHI | BAF | BAT | RHI |
| Normal | 3 | 3 | 5 | 3 | 3 | 5 | 3 | 3 | 5 |
| Periodic waterlogged | 3 | 3 | 5 | 3 | 3 | 5 | 3 | 3 | 5 |

Note: TP = tillering, GP = grand growth, RP = ripening.

BAF = bulk around the field, BAT = bulk around the tree, RHI = rhizosphere.

from 15 positions were mixed equally to obtain a composite sample of 500 g, which was then submitted for chemical property analysis at the Department of Soil Science, Faculty of Agriculture, Kasetsart University, Thailand. The soil samples underwent a series of preparation steps: they were first air-dried, then gently crushed using an agate mortar and pestle. Following this, they were passed through a 2-mm stainless steel sieve and homogenized before analysis. These prepared samples were used to measure soil chemical properties. For soil organic carbon and total nitrogen content analysis, soil samples were finely crushed and sifted through a 0.5 mm sieve, after which they were stored in plastic bags. Soil pH was measured by a pH meter using a soil-per-water ratio of 1:1 [25]. The percentage of soil organic carbon was determined by the Walkley and Black Titration method [26]. Available phosphorus was measured using the Bray II protocol [27]. Potassium, calcium, and magnesium were extracted from the soil with 1N ammonium acetate ($NH_4OAc$) and measured by the atomic absorption spectrophotometer [28,29]. Kjeldahl nitrogen determination was used for total nitrogen measurement [30]. Total phosphorus was determined by the perchloric acid ($HClO_4$) digestion method [31]. The amount of $NO_3^- - N$ and $NH_4^+ - N$ was analyzed by steam distillation with magnesium oxide (MgO) and Devarda's alloy, following the method described by Keeney and Nelson (1982) [32].

## DNA extraction and 16S rRNA and ITS amplicon sequencing

DNA samples were extracted using the DNeasy PowerSoil Pro Kit (Qiagen, Hilden, Germany) according to the manufacturer's instructions. The DNA was further purified using Agencourt AMPure XP beads (Beckman Coulter), and the concentration and quality of DNA were assessed using a Nanodrop Photometer NP-80 (IMPLEN, Westlake Village, CA, USA) by measuring the absorbance ratios at OD 260/280 and OD 260/230, respectively. The genomic DNAs were submitted to Genome Quebec Innovation Centre, McGill University, Canada for 16S and ITS amplicon sequencing. Forward primer 515F (5'-`GTGCCAGCMGCCGCGGT AA`-3') and reverse primer 806R (5'-`GGACTACHVGGGTWTCTAAT`-3') were used for the hypervariable region V4 of 16S rRNA gene amplification specifically for bacteria and archaea. Forward primer ITS1F (5'-`CTTGGTCATTTAGAGGAAGTAA`-3') and reverse primer ITS2R (5'-`GCTGCGTTCTTCATCGATGC`-3') were used for the internal transcribed spacer 1 (ITS1) amplification for fungi. Both 16S and ITS samples were sequenced using Illumina MiSeq (250 bp) and NovaSeq (251 bp) paired-end sequencing platforms (S1 Table). The raw reads have been deposited at the NCBI repository under BioProject accession number PRJNA658446 and SRA accession numbers SRR14677846—SRR14679243.

## Microbiome data analysis

Microbiome data analysis was performed using QIIME2 pipeline v2020.11 [33]. In brief, raw reads were imported into QIIME2 and the quality of raw reads was assessed. Adapters and primers were removed using cutadapt (QIIME2 plugin) and denoised using DADA2 (QIIME2 plugin) with parameters: 16S rRNA—p-trunc-len-f 227 and—p-trunc-len-r 226 and ITS—p-trunc-len-f 224 and—p-trunc-len-r 196. The preprocessed reads were dereplicated into amplicon sequence variants (ASVs). A model of V4 regions was trained using 515F and 806R primers against SILVA SSU database release 132 [34]. ASVs of 16S rRNA were taxonomically classified based on the V4 model. The UNITE database version 8.2 [35] was used to assign fungal ITS ASVs. The raw abundance tables were normalized using the Geometric Mean of Pairwise Ratios (GMPR) [36]. Shannon, Chao1, and Simpson diversity indices were calculated and then statistically tested according to types of soils, stages, and fields as explanatory factors by t-test (P-values < 0.05). Principal coordinate analysis (PCoA) was performed based on Bray-Curtis dissimilarity using the MicrobiomeAnalyst platform [37,38]. Differential abundance

analysis was executed to investigate overrepresented taxa between normal and periodic water-logged fields in each sugarcane growth phase at the genus level using LEfSe version 1.1 [39] on the Galaxy web platform.

## Microbial co-occurrence network analysis

Microbial networks of different soil compartments from normal (N) and periodic waterlogged (W) conditions were separately constructed, including N-BAF, N-BAT, N-RHI, W-BAF, W-BAT, and W-RHI networks. Taxa with members of less than 10 reads per group and with zero counts more than or equal to 5 out of 9 samples (8 out of 15 samples for N-RHI and W-RHI) were removed from the raw abundance table. SparCC version 1.10.1 [40] was used to calculate associations between microbial taxa along selected samples with default settings, and two-sided pseudo P-values were calculated using 100 simulated datasets. Adjusted P-values were estimated using Bonferroni correction [41]. The significant interactions among microbe-microbe were determined as significant correlations with P-values less than 0.05 and absolute correlations of more than 0.8. Microbial networks were visualized using Cytoscape version 3.8 [42]. Network properties including node degree, edge number, clustering coefficient, average number of neighbors, network diameter, and network density were analyzed with Network Analyzer version 4.4.6 [43]. The microbial hubs were then identified based on the following criteria: i) an average relative abundance of three soil compartments with greater than 1%, ii) node degree more than 10, and iii) betweenness centrality less than 0.5.

## Results

### Growth characteristics of sugarcane and soil properties

The aisle between the normal and periodic waterlogged fields spanned approximately 500 m. Both fields adhered to the same farming practices; however, distinctions in sugarcane growth characteristics were noted during the ripening phase (at 9 months). The growth characteristics of sugarcane from both fields were compiled and are presented in the S2 Table. As indicated in Table 2, a t-test analysis revealed that stalk height and internode number per stalk in the normal field were significantly higher than those in the periodic waterlogged field.

The soil pH in the periodic waterlogged field showed a moderate acidic pH of 5.7 whereas in the normal field, a neutral pH of 6.6 was detected (Table 3). Available potassium, magnesium, nitrate nitrogen, ammonium nitrogen, and electrical conductivity (EC) in the periodic waterlogged field were higher than those detected in the normal field (Table 3). In contrast, soil organic matter, available phosphorus, and calcium in the normal field were higher than those in the periodic waterlogged field (Table 3).

**Table 2. Growth characteristics of sugarcane at ripening stage in normal and periodic waterlogged fields.**

|  | Stalk height (cm) | Stalk diameter (cm) | Stalk number per stool | Internode length (cm) | Internode number per stalk |
|---|---|---|---|---|---|
| **Normal** | 288.59 ± 24.12 | 4.95 ± 0.43 | 7.67 ± 0.58 | 11.74 ± 1.48 | 29.88 ± 4.50 |
| **Waterlogging** | 213.59 ± 44.05 | 4.84 ± 0.54 | 4.67 ± 0.58 | 11.36 ± 1.99 | 21.24 ± 3.70 |
| **T-test** | 8.8e-08*** | 0.5 ns | 0.23 ns | 0.51 ns | 3.8e-06*** |

Note:

***: P < 0.001

**: P < 0.01

*: P < 0.05; NS: Not significant.

**Table 3. Soil chemical properties in both normal and periodic waterlogged fields at the ripening stage.**

| Soil chemical property | Normal field | Periodic waterlogged field |
|---|---|---|
| Soil pH | 6.6 | 5.7 |
| Soil organic matter | 2.78 | 1.74 |
| Total N (%) | 0.09 | 0.09 |
| Total P (%) | 0.024 | 0.025 |
| Available P (mg kg$^{-1}$) | 3.44 | 2.68 |
| Available K (mg kg$^{-1}$) | 51 | 88 |
| Ca (mg kg$^{-1}$) | 2.176 | 2.052 |
| Mg (mg kg$^{-1}$) | 268 | 356 |
| NO$^-_3$-N (mg kg$^{-1}$) | 8.4 | 9.8 |
| NH$^+_4$-N (mg kg$^{-1}$) | 1.39 | 4.19 |
| Electrical conductivity (μs m$^{-1}$) | 30.2 | 37.7 |

## Microbiome profile and diversity analysis

A total of 22,082,840 and 15,612,641 raw reads of 16S rRNA sequence (V4 region) and ITS sequence were obtained from all samples of both normal and periodic waterlogged sugarcane plantations, respectively. After preprocessing, 16,098,886 high-quality 16S rRNA sequences and 11,835,650 high-quality ITS sequences were used for downstream analysis. The total features of the former and the latter were 51,074 and 9,840, respectively. After normalization with GMPR, these sequences were classified with more than 99% identity into 58 phyla, 195 classes, 467 orders, 775 families, and 1,644 genera. The top 10 most abundant phyla which accounting for 94.44% were Acidobacteria, Actinobacteria, Armatimonadetes, Chloroflexi, Firmicutes, Gemmatimonadetes, Planctomycetes, Proteobacteria, Rokubacteria, and Verrucomicrobia (Fig 1A). Actinobacteria was the major phylum observed in all samples with 27.04% relative abundance (S3 Table). *Streptomyces*, a well-known antibiotic producer in Actinobacteria [44], was observed in all sample types with > 1% relative abundance. Proteobacteria and Verrucomicrobia had higher proportions in the Rhizosphere than other soil compartments. Rokubacteria tended to be found more in the normal field than periodic waterlogged. In terms of the fungal community, the most abundant phyla consisted of Ascomycota, Basidiomycota, Calcarisporiellomycota, Chytridiomycota, Glomeromycota, Kickxellomycota, Mortierellomycota, and Rozellomycota (Fig 1B). Among them, the majority of fungal phyla across samples was Ascomycota (average of 79.75% relative abundance: S4 Table). In addition, Basidiomycota showed a higher abundance in periodic waterlogged than in the normal field.

The t-test was applied to measure the statistically significant differences in bacterial (Fig 2A–2C) and fungal (Fig 2D–2F) diversities among all soil compartments collected from sugarcane fields. Specifically, Shannon's, Chao1, and Simpson's diversity indices were calculated for each sample using the vegan R package. In the bacterial community, all samples had Shannon's diversity index of more than 5.0, Chao1 diversity index of more than 500, and Simpson's diversity index of more than 0.9. In contrast, the fungal community had lower Shannon's and Chao1 diversity indices than the bacteria, ranging from 1.0 to 4.0 and 108 to 362, respectively. Moreover, Simpson's diversity index varied and ranged from 0.4 to 0.9. The statistical tests of 16S rRNA sequences revealed that alpha diversities of normal and periodic waterlogged fields were significantly different in GP-BAF and GP-RHI according to Shannon's index; GP-BAF and RP-BAT according to the Chao1 index; and GP-BAT, RP-BAT, TP-RHI, GP-RHI, and RP-RHI according to Simpson's index. For ITS data, TP-BAF and GP-BAF were significantly different based on the Chao1 diversity index.

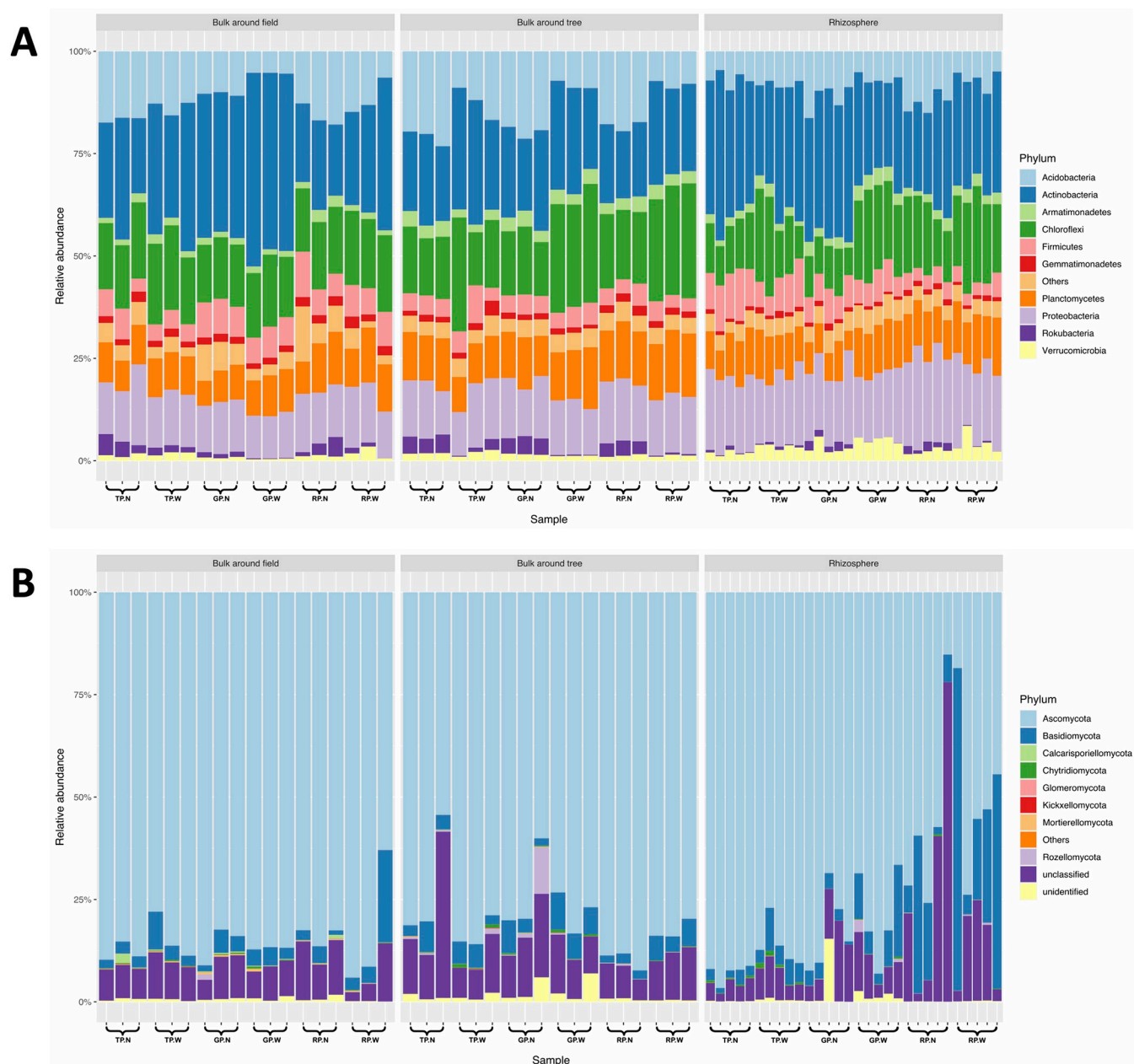

**Fig 1.** Taxonomic composition of (A) bacterial and (B) fungal communities at the phylum level. N and W after TP, GP, and RP represented normal and periodic waterlogged fields, respectively.

The Bray-Curtis dissimilarity was computed among all samples in each developmental phase, visualized using principal coordinate analysis (PCoA), and clustered into normal and periodic waterlogged conditions (Fig 3). According to the PCoA result, the communities of bacteria in the sugarcane grand growth phase (GP) in the normal plantation field were clearly separated from the communities reported in periodic waterlogged samples (Fig 3B). However, the permutational multivariate analysis of variance (PERMANOVA) test for assessing the strength and statistical significance of sample clustering showed significant differences in bacterial communities in all growth phases with $P < 0.05$ (S5 Table). For fungal communities,

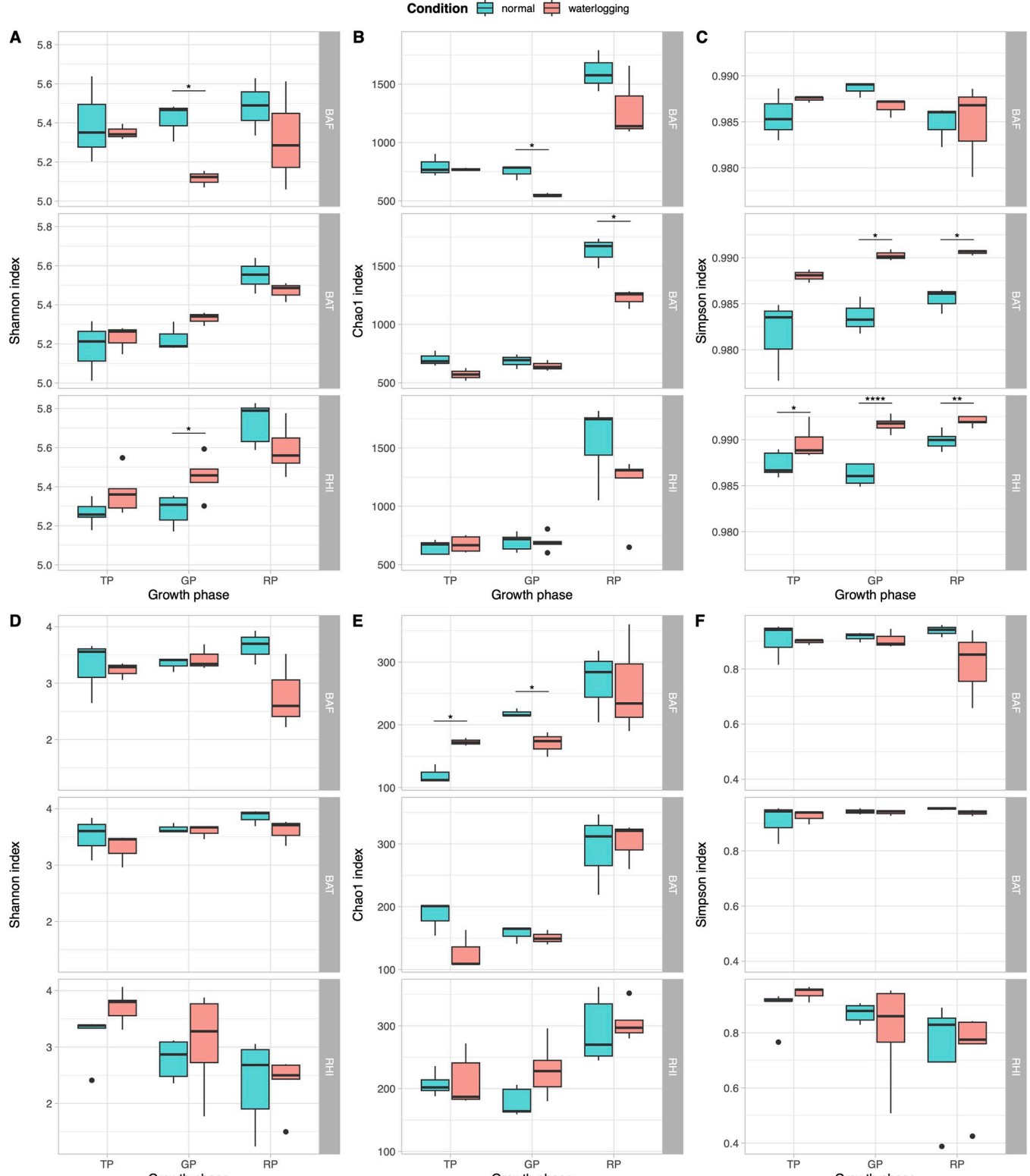

**Fig 2.** Boxplots demonstrate the distributions of alpha diversities of (A-C) bacterial (16S rRNA) and (D-F) fungal (ITS) communities using Shannon's, Chao1, and Simpson's diversity indices in each developmental phase.

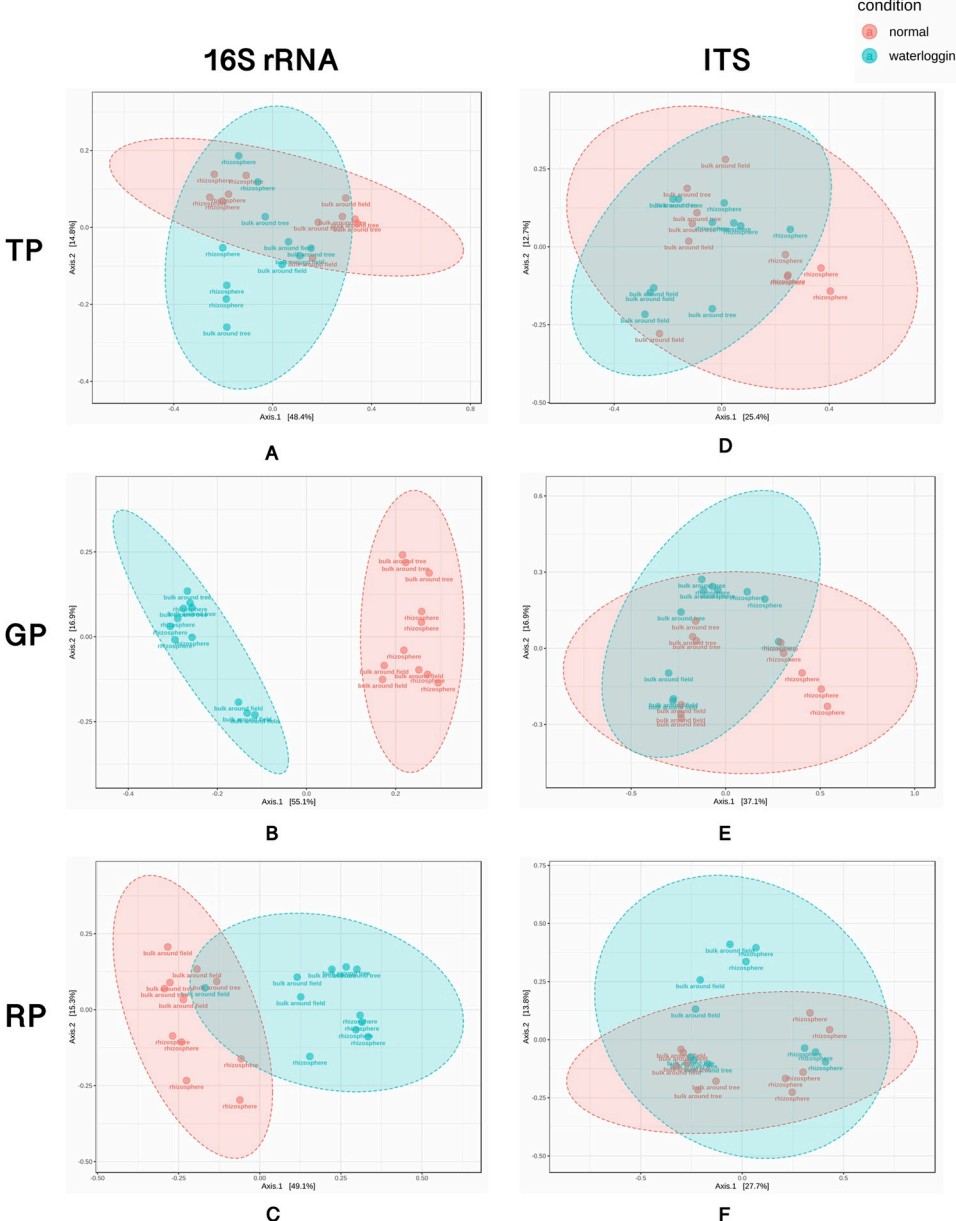

**Fig 3.** PCoA plots based on the Bray-Curtis dissimilarity matrix indicate beta diversities of normal and periodic waterlogged samples in all developmental phases (TP, GP, and RP) in (A-C) bacterial community using 16S rRNA and (D-F) fungal community with ITS.

PERMANOVA analysis revealed statistically significant differences in sample clustering between the TP and GP stages, while in RP, there were no significant differences among the samples. For bacterial communities, there was a weaker association in RP in both fields (S5 Table).

## Identification of possible microbial markers for sugarcane fields

To identify microbial or bacterial genera that were specifically enriched in either normal or periodic waterlogged fields in different growth phases and soil compartments, LEfSe analysis

**Table 4. The number of differentially abundant taxa of bacteria, archaea, and fungi in normal and periodic waterlogged fields.**

| | | Number of differential microbial abundances in normal field | Number of differential microbial abundances in periodic waterlogged field | Total number of microbial differential abundances |
|---|---|---|---|---|
| TP | BAF | 24 / 5 | 20 / 8 | 44 / 13 |
| | BAT | 42 / 22 | 29 / 7 | 71 / 29 |
| | RHI | 12 / 3 | 15 / 16 | 27 / 19 |
| GP | BAF | 83 / 19 | 48 / 17 | 131 / 36 |
| | BAT | 72 / 19 | 53 / 11 | 125 / 30 |
| | RHI | 79 / 10 | 78 / 38 | 157 / 48 |
| RP | BAF | 42 / 26 | 43 / 5 | 85 / 31 |
| | BAT | 120 / 18 | 68 / 18 | 188 / 36 |
| | RHI | 130 / 9 | 73 / 22 | 203 / 31 |

Note: Values in the table represent the abundances of bacteria and archaea/fungi.

was applied with an LDA score > 2 as the cut-off. Table 4 showed that RP had the highest number of differential taxa among bacterial genera, while TP had the lowest number of differential taxa, across different growth phases. In terms of the soil compartment, the highest number of microbial genera was reported in RHI with 203 and 157 microbial genera in RP and GP, respectively (Table 4). BAT possessed a greater number of unique genera (71) in TP, while BAF had the median number of specific taxa in TP and GP and the minimum in RP (Table 4). Furthermore, the normal field had a greater number of unique microbes than the periodic waterlogged field (Table 4). Genus RB41 was detected as the most differential taxon in GP-BAT, TP-BAT, and RP-RHI of the normal field with LDA scores of 4.79, 4.75, and 4.30, respectively (S6 Table). *Piscinibacter*, *Streptomyces*, *Bacillus*, *Geodermatophilus*, *Singulisphaera*, and *Conexibacter* were the most differential genera in RP-BAT (LDA score = 4.68; normal field), GP-RHI (LDA score = 4.54; normal field), TP-RHI (LDA score = 4.32; normal field), GP-BAF (LDA score = 4.21; periodic waterlogged field), RP-BAF (LDA score = 4.09; periodic waterlogged field), and TP-BAF (LDA score = 4.07; periodic waterlogged field), respectively (S6 Table).

For fungal genera, the highest number of unique taxa was reported in GP, followed by RP and TP (Table 4). In particular, RHI possessed the most abundant specific genera (48) in GP, while BAF possessed the fewest specific genera in TP with 13 taxa (Table 4). Furthermore, the RHI of the periodic waterlogged field had a higher number of enriched taxa compared to the normal field in all the phases (Table 3). *Penicillium* was the fungal genus with the highest differential abundance, with an LDA score of 5.03 in TP-RHI of the normal field, followed by *Aspergillus* in TP-BAF (LDA score = 4.98; periodic waterlogged field), *Hamigera* in RP-BAF (LDA score = 4.96; periodic waterlogged field), *Acremonium* in GP-RHI (LDA score = 4.88; normal field), *Humicola* in GP-BAF (LDA score = 4.86; normal field), and *Talaromyces* in TP-BAT (LDA score = 4.63; normal field) (S7 Table). Furthermore, *Scytalidium* was at the top of unique fungi in GP-BAT and RP-BAT of the periodic waterlogged field with LDA scores of 4.41 and 4.34, respectively. *Conioscypha* had the highest LDA score of 4.33 in RP-RHI of the periodic waterlogged field (S7 Table).

## Soil microbial network analysis

A soil microbial network was constructed for each soil compartment and condition (N-BAF, N-BAT, N-RHI, W-BAF, W-BAT, and W-RHI) based on the correlation analysis. SparCC [40] was applied to calculate the correlations between these two genera in both normal and periodic

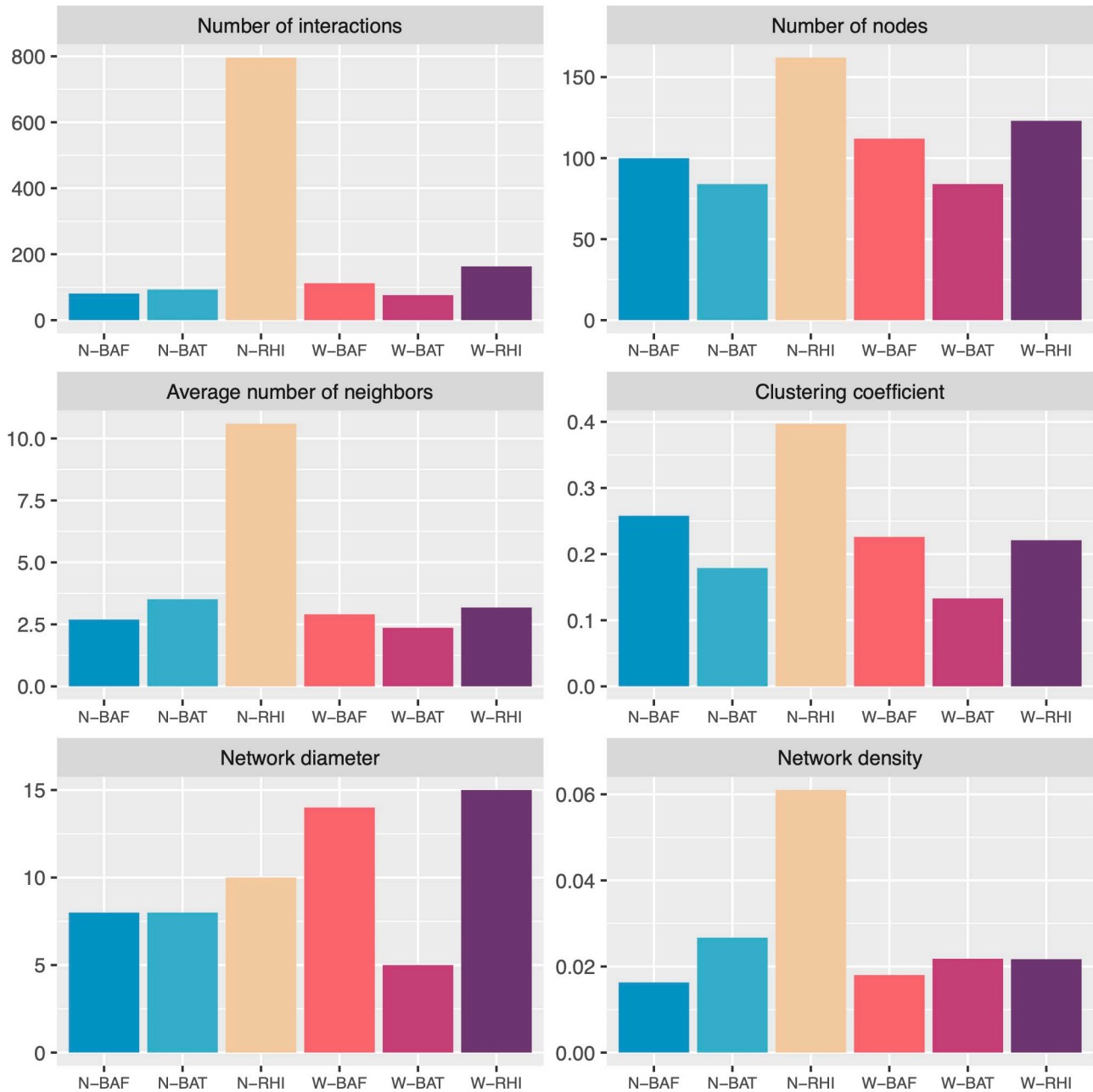

**Fig 4. Network characteristics included the number of interactions, number of nodes, average number of neighbors, clustering coefficient, network diameter, and network density of N-BAF, N-BAT, N-RHI, W-BAF, W-BAT, and W-RHI networks.**

waterlogged fields. Significant relationships with |r| > 0.8 and adjusted P-value < 0.05 according to Bonferroni correction were selected for microbial network reconstruction (S1 Fig). To evaluate these networks, their characteristics were measured in terms of the number of interactions, number of nodes, clustering coefficient, average number of neighbors, network diameter, and network density (Fig 4). N-RHI revealed the highest value in five metrics, including (i) 796 interactions (539 positive and 257 negative interactions), (ii) 162 nodes (1 archaeon, 123 bacteria, and 38 fungi), (iii) an average number of neighbors of 10.591, (iv) a clustering coefficient of 0.397, and (v) a network density of 0.061, indicating the greatest connection and

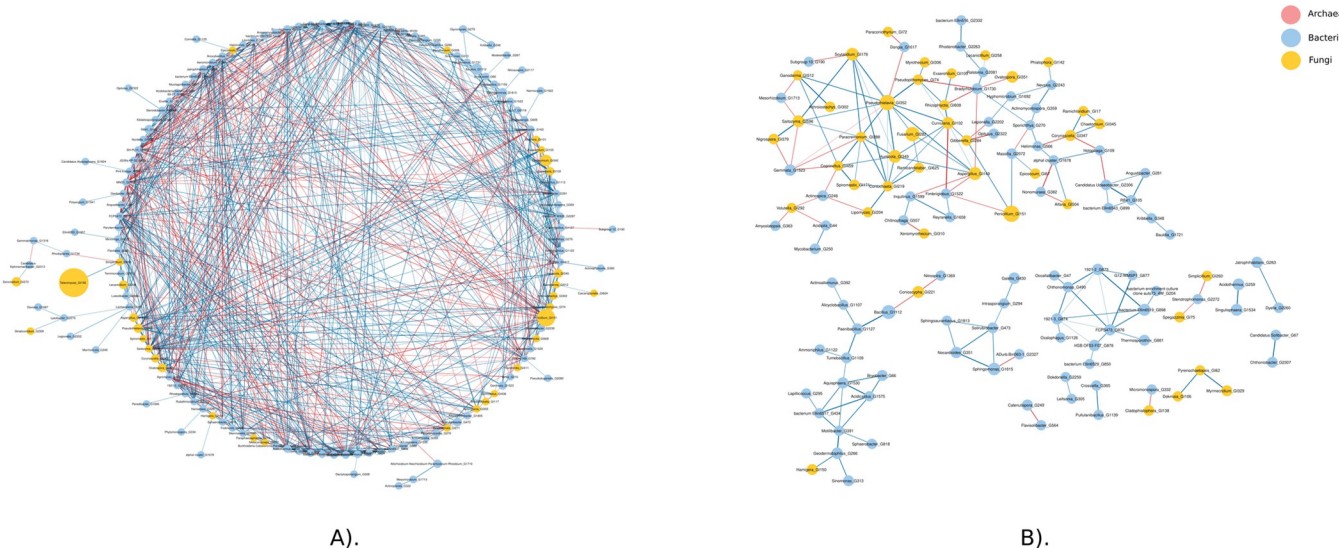

A).                                                                        B).

**Fig 5.** Rhizosphere microbial networks were constructed using SparCC with |r| > 0.8 and adjusted P-value < 0.05, including A) N-RHI, and B) W-RHI. Node color represents the domain of the organisms (pink: Archaea, blue: Bacteria, and yellow: Fungi). Size of the node corresponds to the average relative abundance of three growth stages (TP, GP, and RP). Blue and red edges are positive and negative correlations, respectively. Intensity of edge color decodes a range of correlations. Nodes with black borders refer to hubs.

complexity among all six networks (Fig 4). The microbial community in the normal condition tended to have higher complexity networks than that of the periodic waterlogged condition based on network characteristics. According to the comparison of microbial networks between three soil compartments (BAF, BAT, and RHI), rhizosphere networks were found to have the largest number of microbes with correlated interactions. Moreover, rhizosphere microbes tended to form clusters with high density in both the normal and periodic waterlogged fields, as reflected by the clustering coefficient shown in Fig 5.

Microbial hubs were identified for each network based on the following criteria: node degree >10, betweenness centrality <0.5, and average percent relative abundance >1. Interestingly, 13 hubs consisting of 7 fungal genera (*Penicillium*, *Pseudothielavia*, *Aspergillus*, *Humicola*, *Curvularia*, *Saitozyma*, and *Ovatospora*) and 5 bacterial genera (*Sphingomonas*, *Singulisphaera*, *Ellin6519*, *Acidothermus*, and *Conexibacter*), were found in N-RHI, whereas only one fungal genus (*Pseudothielavia*) hub was observed in W-RHI (S8 Table). Moreover, hub degrees of N-RHI were higher than those of W-RHI, whereas no hub was found in other networks.

## Discussion

In the present study, the amplicon sequencing approach based on 16S rRNA and ITS marker genes was employed to determine the impact of periodic waterlogged stress on the diversity and composition of rhizosphere and soil bacterial and fungal communities along the growth phase of sugarcane. Periodic waterlogging could reduce oxygen levels, elevate denitrification, increase volatile acid, and decrease pH in the soil, which can result in shifts in soil microbial composition [45,46] and reduced sugarcane growth and productivity [9]. Differential abundance analysis was carried out to investigate candidate microbial markers for improving soil quality. Co-occurrence network analysis was performed to reveal the complexity and connectivity of microbial communities influenced by periodic waterlogged stress.

Based on 16S rRNA sequencing analysis, the phylum of Acidobacteria was decreased in the periodic waterlogged condition across all growth stages, while Actinobacteria and Chloroflexi

were elevated. Actinobacteria (27.04% relative abundance), commonly known as plant growth-promoting rhizobacteria (PGPR) [47], was the most dominant phylum in both normal and periodic waterlogged fields (S3 Table). Anaerolineaceae and Ktedonobacteraceae, which belonged to Chloroflexi, have been reported to be the predominant groups in soils with low pH and oxygen levels [48], and likely to inhabit low to moderate-pH soil [49], respectively. Acidobacteria, the prevalent phylum in sugarcane soil, can competitively colonize plant rhizosphere and establish beneficial relationships with plants. They are also involved in biogeochemical cycles, decomposition of biopolymers, secretion of exopolysaccharides, and plant growth-promoting activities [4]. The reduction of Acidobacteria in periodic waterlogged soil could imply lower growth of sugarcane through these interactions. Based on the soil fungal taxonomic assignment, the most abundant phylum in all samples was Ascomycota (79.75%). In periodic waterlogged condition, Ascomycota was slightly decreased while Basidiomycota was elevated. Ascomycota is the largest phylum of fungi and contains a wide range of plant growth-promoting fungi (PGPF) such as *Aspergillus*, *Chaetomium*, *Cladosporium*, *Exophiala*, *Penicillium*, *Trichoderma*, *Phoma*, *Phomopsis*, *Purpureocillium*, and *Talaromyces* for various plant types, from arabidopsis and flower plants to vegetables like cucumber and eggplants [50–61]. Among PGPF, *Penicillium*, *Trichoderma*, and *Talaromyces*, which are previously known to secrete important metabolites such as gibberellins, indoleacetic acid, and siderophore to enhance plant growth and alleviate the effects of environmental stressors in wheat and sweet pepper [62,63], were observed in all samples of our study. In the Basidiomycota, *Saitozyma* spp., the unconventional yeast, were enriched in all growth stages of periodic waterlogged field, especially *Saitozyma podzolica* which are typical soil-borne yeasts living in acid soils [64]. The analysis revealed a low amount of possible beneficial fungi while the possible soil-borne pathogen fungi were presented in higher amounts in the periodic waterlogged field when compared with the normal field.

The diversity analysis of soil bacterial and fungal communities showed similar Shannon's diversity indices and Simpson's indices between normal and periodic waterlogged conditions. This result is in accordance with previous studies that found no difference in alpha diversity between drained and flooded agricultural fields [48,65]. However, PERMANOVA analysis revealed significant differences in microbial community structure between these conditions, notably in bacterial communities during various plant growth stages. Differential abundance analysis indicated an increase in bacterial genera from early to late growth stages, with the highest number of specific microbes found in the rhizosphere, supporting the fact that the rhizosphere is the area where plants actively excrete root exudates to selectively attract specific microbes to support their growth and development [66]. In normal condition, well-known plant growth-promoting bacteria (PGPB) were enriched, including *Arthrobacter*, *Azoarcus*, *Bacillus*, *Paenibacillus*, *Pseudomonas*, and *Streptomyces*. In contrast, phytopathogens such as bacterium *Ralstonia* and fungi *Colletotrichum*, *Fusarium*, and *Volutella*, and bacteria causing the low efficiency of N fertilizer application in agriculture like ammonia oxidizing *Nitrosococcus* and denitrifying *Conexibacter* were identified as potential biomarkers in the periodic waterlogged field. Furthermore, periodic waterlogged stress has been known to decrease soil pH and oxygen levels [67], thus it is not surprising that this area was dominated by acidophilic and anaerobic bacteria, including *Acidibacter*, *Acidicaldus*, *Acidiphilum*, *Acidisoma*, *Acidobacteria bacterium* LWQ13, *Anaerolinea*, *Longilinea*, and *Acidothermus*. In addition, *Asticcacaulis*, which is known to be a plant growth-inhibiting bacterium [68], was also enriched in the periodic waterlogged field. In contrast, several plant growth-promoting fungi (PGPF), such as *Chaetomium* spp. which play an important role in promoting plant growth by providing them with resistance against several stresses [69–74], enriched in normal condition. Remarkably, *C. elatum* was linked to the production of chaetoglobosin-C, known for its antifungal properties against *Fusarium*, which can be pathogenic to various plants, including tomatoes and

sugarcane [52,75,76]. Notably, some PGPF were detected under periodic waterlogged stress. For example, *P. lilacinum*, a member of *Purpureocillium*, this species was found in all soil compartments and stages of sugarcane, aligning with other studies that have frequently identified it in the rhizosphere of various crops for its biocontrol potential against destructive root-knot nematodes in maize, beans, and soybeans [77,78].

Moreover, co-occurrence network analysis revealed that the normal condition exhibited greater microbial complexity, especially in the rhizosphere, where microbial interactions and niche sharing were most pronounced. The normal rhizosphere network displayed the highest connectivity, suggesting enhanced microbial communication, potentially facilitating plant responses to both abiotic and biotic stresses, and consequently increasing soil biological fertility [79–81]. Importantly, the result suggested that enrichment of these pathogenic, ammonia-oxidizing, and denitrifying microbes might be associated with the decline of sugarcane growth in the periodic waterlogged field.

Importantly, our study revealed that periodic waterlogged stress had a long-term adverse effect on the soil microbial community, resulting in a reduction of sugarcane growth and development. The shift in microbial functions was closely linked to plant growth and development. In the normal field, the microbiome profiling revealed a range of beneficial microbes, including nitrogen and carbon dioxide-fixing bacteria (*Desulfitobacterium*), phosphatase and beta-glucosidase-producing bacteria (*Amnibacterium kyonggiense*), fungal plant pathogens-inhibiting microbes (*Streptomyces* and *Chaetomium*), and plant growth hormones-producing microbes (*Promicromonospora*, *Penicillium*, *Trichoderma*, and *Talaromyces*) [75,82–86]. Conversely, the analysis revealed a multitude of microbes with negative impacts on sugarcane growth in the periodic waterlogged field. These microbes are related to fungal plant pathogens (*Colletotrichum* and *Fusarium*), ammonia-oxidizing bacteria (*Nitrosocosmicus* and *Rhodopirellula*), denitrifying bacteria (*Conexibacter*), and plant growth-inhibiting bacteria (*Asticcacaulis*) [87–89]. The overall results provided insights into the relationships among microbial species in the periodic sugarcane waterlogged field. The results could be applied as guidelines for constructing microbial communities for soil amendment in sugarcane improvement, especially in waterlogged or deteriorated agricultural areas. Furthermore, the knowledge gained from this research holds promise for future applications in precision agriculture management strategies.

## Conclusion

This study demonstrated that the periodic waterlogged condition in the field crucially affects sugarcane growth, soil microbial structures, and their interactions. Several PGPM were identified as potential biomarkers for promoting sugarcane growth in the normal field, including *Paenibacillus*, *Pseudomonas*, *Streptomyces*, and *Chaetomium*. Besides, fungal plant pathogens, ammonia-oxidizing bacteria, denitrifying bacteria, and plant growth-inhibiting bacteria, were identified as potential biomarkers of the periodic waterlogged area. Co-occurrence network analysis suggested that in the normal field, the interactions among microbes were more complex than those found in the periodic waterlogged network, which could be linked to plant growth in those areas. Microorganisms inhabiting the rhizosphere had the highest complexity, implying their potential roles in sharing niche resources among microbes and host plants. The microbial hubs found in the normal field rhizosphere and functioning as microbial mediators were mainly PGPF belonging to the Ascomycota phylum. This study, thus, provided an understanding of microbial profiles and interactions under periodic waterlogged conditions in the sugarcane plantation area of Ratchaburi province, Thailand, resulting from a long-term negative effect on sugarcane growth and development. This information can be further applied for sugarcane and soil improvement in sugarcane plantations for sustainable agriculture.

## Supporting information

**S1 Fig.** Co-occurrence networks including (A) N-BAF, (B) N-BAT, (C) W-BAF, and (D) W-BAT. Node color represents domain (pink: archaea, blue: bacteria, and yellow: fungi). Size of the node is the average relative abundance based on three growth stages (TP, GP, and RP). Blue and red edges mean positive and negative correlations, respectively. Intensity of edge color represents a range of correlations.
(DOCX)

**S1 Table. Number of soil replicates in 16S rRNA/ITS in different growth stages (tillering: TP, grand growth: GP, and ripening: RP) and soil compartments (bulk around the field: BAF, bulk around the tree: BAT, and rhizosphere: RHI) in both normal and periodic waterlogged conditions.** These samples were sequenced using Illumina MiSeq platform, except RP which was sequenced using Illumina NovaSeq platform.
(XLSX)

**S2 Table. Growth characteristics of sugarcane cultivated in normal and periodic waterlogged fields.**
(XLSX)

**S3 Table. Relative abundance of archaea and bacteria in normal and periodic waterlogged areas.**
(XLSX)

**S4 Table. Relative abundance of fungi in normal and periodic waterlogged areas.**
(XLSX)

**S5 Table. Tested PERMANOVA result statistics in beta diversity analysis in 16S rRNA and ITS data.**
(XLSX)

**S6 Table. LDA scores for bacteria and archaea based on LEfSe analysis.**
(XLSX)

**S7 Table. LDA scores for fungi based on LEfSe analysis.**
(XLSX)

**S8 Table. Microbial hubs of soil microbial networks.**
(XLSX)

## Acknowledgments

We thank the Systems Biology and Bioinformatics laboratory and Fungal Biotechnology laboratory, KMUTT for the computing resources, and laboratory equipment, respectively. We acknowledge Uncle Pramote for providing soil samples of conventional sugarcane farm for our research.

## Author Contributions

**Conceptualization:** Onnicha Leelastwattanagul, Sawannee Sutheeworapong, Ahmad Nuruddin Khoiri, Sudarat Dulsawat, Songsak Wattanachaisaereekul, Anuwat Tachaleat, Thanawat Duangfoo, Peerada Prommeenate, Supapon Cheevadhanarak, Jiraporn Jirakkakul.

**Data curation:** Onnicha Leelastwattanagul, Sawannee Sutheeworapong, Ahmad Nuruddin Khoiri.

**Formal analysis:** Onnicha Leelastwattanagul, Sawannee Sutheeworapong, Ahmad Nuruddin Khoiri, Anuwat Tachaleat, Supapon Cheevadhanarak.

**Funding acquisition:** Sawannee Sutheeworapong, Songsak Wattanachaisaereekul, Anuwat Tachaleat, Peerada Prommeenate, Supapon Cheevadhanarak, Jiraporn Jirakkakul.

**Investigation:** Onnicha Leelastwattanagul, Sawannee Sutheeworapong, Ahmad Nuruddin Khoiri, Sudarat Dulsawat, Songsak Wattanachaisaereekul, Anuwat Tachaleat, Thanawat Duangfoo, Peerada Prommeenate, Jiraporn Jirakkakul.

**Methodology:** Onnicha Leelastwattanagul, Sawannee Sutheeworapong, Ahmad Nuruddin Khoiri, Sudarat Dulsawat, Songsak Wattanachaisaereekul, Anuwat Tachaleat, Thanawat Duangfoo, Prasobsook Paenkaew, Peerada Prommeenate, Supapon Cheevadhanarak, Jiraporn Jirakkakul.

**Project administration:** Onnicha Leelastwattanagul, Sawannee Sutheeworapong, Songsak Wattanachaisaereekul, Peerada Prommeenate, Jiraporn Jirakkakul.

**Resources:** Onnicha Leelastwattanagul, Sawannee Sutheeworapong, Ahmad Nuruddin Khoiri, Sudarat Dulsawat, Songsak Wattanachaisaereekul, Anuwat Tachaleat, Thanawat Duangfoo, Peerada Prommeenate, Jiraporn Jirakkakul.

**Software:** Onnicha Leelastwattanagul, Sawannee Sutheeworapong, Ahmad Nuruddin Khoiri, Prasobsook Paenkaew.

**Validation:** Onnicha Leelastwattanagul, Sawannee Sutheeworapong, Ahmad Nuruddin Khoiri, Sudarat Dulsawat, Songsak Wattanachaisaereekul, Thanawat Duangfoo.

**Visualization:** Onnicha Leelastwattanagul, Ahmad Nuruddin Khoiri.

**Writing – original draft:** Onnicha Leelastwattanagul, Sawannee Sutheeworapong, Ahmad Nuruddin Khoiri, Sudarat Dulsawat, Songsak Wattanachaisaereekul, Thanawat Duangfoo, Jiraporn Jirakkakul.

**Writing – review & editing:** Onnicha Leelastwattanagul, Sawannee Sutheeworapong, Ahmad Nuruddin Khoiri, Sudarat Dulsawat, Songsak Wattanachaisaereekul, Anuwat Tachaleat, Thanawat Duangfoo, Prasobsook Paenkaew, Peerada Prommeenate, Supapon Cheevadhanarak, Jiraporn Jirakkakul.

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
