## [Decision Letter · Decision Letter 0]

10 Jul 2023

PONE-D-23-16598Soil Microbiome Analysis Reveals Effects of Period Waterlogging Stress on Sugarcane GrowthPLOS ONE

Dear Dr. Jirakkakul,

Thank you for submitting your manuscript to PLOS ONE. After careful consideration, we feel that it has merit but does not fully meet PLOS ONE’s publication criteria as it currently stands. Therefore, we invite you to submit a revised version of the manuscript that addresses the points raised during the review process.

We look forward to receiving your revised manuscript.

Kind regards,

Erika Kothe

Academic Editor

PLOS ONE

Journal Requirements:

"This research project is supported by Thailand Science Research and Innovation (TSRI) Basic Research Fund: Fiscal year 2023 under project number: 181932."

Additional Editor Comments:

The reviewers identified several essential shortcomings, specifically full description of soil daran and complete soil management needs to be included in materials and methods. This might then even allow for a CCA analysis. If the authors feel they can supply that in addition to amending all other suggestions by the reviewers, uploading a major revision is possible. If time is too short for that, the authors should not go revision, but instead submit a new publication when a full dataset is available.

Reviewers' comments:

Reviewer's Responses to Questions

**Comments to the Author**

1. Is the manuscript technically sound, and do the data support the conclusions?

Reviewer #1: Yes

Reviewer #2: Yes

2. Has the statistical analysis been performed appropriately and rigorously? 

Reviewer #1: Yes

Reviewer #2: No

3. Have the authors made all data underlying the findings in their manuscript fully available?

Reviewer #1: Yes

Reviewer #2: Yes

4. Is the manuscript presented in an intelligible fashion and written in standard English?

Reviewer #1: Yes

Reviewer #2: Yes

5. Review Comments to the Author

Reviewer #1: The draft titled “Soil Microbiome Analysis Reveals Effects of Period Waterlogging Stress on Sugarcane Growth”. The idea of experimental design is novel, and the work is meaningful. However, as a journal article to be published, there are many deficiencies.

Results: The results can be further simplify. The figures look uncomfortable, especially figure 5 about microbial network suggest visualization by cytoscape or gephi. And the figure 1 should reflecting the abundance of archeobacteria. As for Figures 2 and 3, it is also recommended to include analysis of archeobacteria.

Discussions: This should have a thorough discussion with the results of previous studies, not showcasing the results.

Reviewer #2: The overall structure of the article is complete and the content is substantial, but there are still some small problems that need to be added.

Line 24: The abstract need to be further condensed.

Line 39-41: These contents are not the results of this paper.

Line 52: The Latin name for sugarcane needs to be supplemented, please refer to the study of Xiao et al. (2023) (https://doi.org/10.1016/j.catena.2023.107000) and Xiao et al. (2023) (https://doi.org/10.1016/j.eti.2023.103244)

Line 87: Other studies or results of periodic waterlogging on sugarcane plants or soils should be mentioned.

Line 104: The GPS coordinates (eg. latitude and longitude) of the test site, soil type and state, altitude and other relevant information about the test site need to be supplemented

Line 107-111: Sugarcane planting information, fertilization, irrigation and field management information all need to be supplemented, please refer to the study of Yang et al. (2023) (https://doi.org/10.1016/j.apsoil.2022.104699)

Line 151: Raw data from 16s and ITS sequencing should be uploaded to relevant databases such as National Center for Biotechnology Information (NCBI).

Line 162-164: As far as we know, the Shannon and Simpson indices are used to indicate microbial diversity, why not analyze microbial abundance (generally indicated by Chao1 and Ace indices)?

Line 164: According to the format requirements of the journal, “p” should be rewritten as “P”.

Line 190-191: This sentence should be moved to the materials and methods section.

Line 192-194: Has the data in Table 2 been analyzed for significance? Why the relevant marks and descriptions are not showed in Table 2 and table notes?

Line 206: Do these soil chemical properties differ significantly between the normal field and the periodic waterlogged field?

6. PLOS authors have the option to publish the peer review history of their article (what does this mean?). If published, this will include your full peer review and any attached files.

Reviewer #1: No

Reviewer #2: No

---

## [Author Response · Author response to Decision Letter 0]

24 Aug 2023

Response to the Reviewer #1

Reviewer #1: 

1. The draft titled “Soil Microbiome Analysis Reveals Effects of Period Waterlogging Stress on Sugarcane Growth”. The idea of experimental design is novel, and the work is meaningful. However, as a journal article to be published, there are many deficiencies.

ANS: 

The title should be revised from “Soil Microbiome Analysis Reveals Effects of Period Waterlogging Stress on Sugarcane Growth” to “Soil Microbiome Analysis Reveals Effects of Periodic Waterlogging Stress on Sugarcane Growth”.

2. Results: The results can be further simplify. 

The figures look uncomfortable, especially figure 5 about microbial network suggest visualization by cytoscape or gephi. 

ANS: 

We adjusted the results and the networks had already been visualized using Cytoscape. We adjusted the colors of the nodes and edges to increase contrast and added the node labels.

3. And the figure 1 should reflecting the abundance of archeobacteria. 

ANS: 

The 16S rRNA gene has the capability to detect both bacteria and archaea. Therefore, our taxonomic abundance analysis considered these two groups collectively. In presenting our results, we aim to underscore the leading ten phyla identified through both the 16S and ITS analyses. Since archaea constituted a minority within the community, they were categorized under 'Others'. Furthermore, other results were analyzed separately based on 16S rRNA and ITS genes.

4. As for Figures 2 and 3, it is also recommended to include analysis of archeobacteria.

ANS: 

From the previous answer, we analyzed separately based on 16S rRNA and ITS genes. So, the archaeal group was included in both diversity analyses.

5. Discussions: This should have a thorough discussion with the results of previous studies, not showcasing the results.

ANS: 

We have adjusted the discussion to be more comprehensive and not just a showcase of the results.

Response to the Reviewer #2

Reviewer #2: 

The overall structure of the article is complete and the content is substantial, but there are still some small problems that need to be added.

1. Line 24: The abstract need to be further condensed.

 ANS:

 The abstract was condensed as per your suggestion.

2. Line 39-41: These contents are not the results of this paper.

ANS:

We have removed the suggested sentences.

3. Line 52: The Latin name for sugarcane needs to be supplemented, please refer to the study of Xiao et al. (2023) (https://doi.org/10.1016/j.catena.2023.107000) and Xiao et al. (2023) (https://doi.org/10.1016/j.eti.2023.103244)

ANS:

We have added the Latin name for sugarcane as Saccharum officinarum L.

4. Line 87: Other studies or results of periodic waterlogging on sugarcane plants or soils should be mentioned.

ANS: 

 We added more details about the effect of waterlogging on sugarcane growth and soil in the introduction part as in lines 73-83.

5. Line 104: The GPS coordinates (eg. latitude and longitude) of the test site, soil type and state, altitude and other relevant information about the test site need to be supplemented.

ANS: 

We have included additional details about the test site in 'The Description of Sugarcane Field Sites' in the 'Materials and Methods' section, specifically in lines 94-99.

6. Line 107-111: Sugarcane planting information, fertilization, irrigation and field management information all need to be supplemented, please refer to the study of Yang et al. (2023) (https://doi.org/10.1016/j.apsoil.2022.104699)

ANS: 

We added more details of fertilization and irrigation information in 'The Description of Sugarcane Field Sites' in the 'Materials and Methods' section, specifically in lines 99-106.

7. Line 151: Raw data from 16s and ITS sequencing should be uploaded to relevant databases such as National Center for Biotechnology Information (NCBI).

ANS:

16S rRNA and ITS gene sequence data associated with this article have been deposited at NCBI under BioProject accession number: PRJNA658446 and SRA accession numbers: SRR14677846 - SRR14679243.

8. Line 162-164: As far as we know, the Shannon and Simpson indices are used to indicate microbial diversity, why not analyze microbial abundance (generally indicated by Chao1 and Ace indices)?

ANS: 

We additionally analyzed the Chao1 index and inserted it in Fig 2.

9. Line 164: According to the format requirements of the journal, “p” should be rewritten as “P”.

ANS:

 We have revised it.

10. Line 190-191: This sentence should be moved to the materials and methods section.

 ANS:

 We have moved it to the Materials and Methods section.

11. Line 192-194: Has the data in Table 2 been analyzed for significance? Why the relevant marks and descriptions are not showed in Table 2 and table notes?

ANS: 

Yes, we have additionally analyzed the significance of sugarcane characteristics between normal and periodic waterlogged fields and provided raw data on sugarcane characteristics in S2 Table. 

12. Line 206: Do these soil chemical properties differ significantly between the normal field and the periodic waterlogged field?

ANS:

We collected soil samples from five points for each field and then pooled them together to measure the soil's chemical properties. Unfortunately, we could not compare it using any statistical test because we did not have replicates for each measurement. Thus, we could not really say whether soil chemical properties between normal and periodic waterlogged fields differ significantly or not.

---

## [Decision Letter · Decision Letter 1]

6 Sep 2023

PONE-D-23-16598R1Soil Microbiome Analysis Reveals Effects of Periodic Waterlogging Stress on Sugarcane GrowthPLOS ONE

Dear Dr. Jirakkakul,

Thank you for submitting your manuscript to PLOS ONE. After careful consideration, we feel that it has merit but does not fully meet PLOS ONE’s publication criteria as it currently stands. Therefore, we invite you to submit a revised version of the manuscript that addresses the points raised during the review process.

Please carefully attend to the additional minor requests made by the reviewer.

We look forward to receiving your revised manuscript.

Kind regards,

Erika Kothe

Academic Editor

PLOS ONE

Journal Requirements:

Reviewers' comments:

Reviewer's Responses to Questions

**Comments to the Author**

1. If the authors have adequately addressed your comments raised in a previous round of review and you feel that this manuscript is now acceptable for publication, you may indicate that here to bypass the “Comments to the Author” section, enter your conflict of interest statement in the “Confidential to Editor” section, and submit your "Accept" recommendation.

Reviewer #2: All comments have been addressed

2. Is the manuscript technically sound, and do the data support the conclusions?

Reviewer #2: Yes

3. Has the statistical analysis been performed appropriately and rigorously? 

Reviewer #2: Yes

4. Have the authors made all data underlying the findings in their manuscript fully available?

Reviewer #2: Yes

5. Is the manuscript presented in an intelligible fashion and written in standard English?

Reviewer #2: Yes

6. Review Comments to the Author

Reviewer #2: The authors have made modifications according to my suggestions before, but there are still some small details in the full text that need to be checked and modified. I hope that the article can be carefully checked and modified before publication.

Line 116-120: How are soils treated and preserved for soil chemical property analysis? Like how many millimeters of sieve?

Line 236-238: References need to be provided.

Line 273, 326, 345: “p” may be rewritten as “P”.

Line 285: “shows” may be rewritten as “showed”.

Line 425: …[80-82]. Importantly, …

7. PLOS authors have the option to publish the peer review history of their article (what does this mean?). If published, this will include your full peer review and any attached files.

Reviewer #2: No

---

## [Author Response · Author response to Decision Letter 1]

18 Oct 2023

Dear editor and Reviewers

We have edited the manuscript as per your suggestions, and we have reviewed the references included in the manuscript.

Comment: Lines 116-120: How are soils treated and preserved for soil chemical property analysis? For example, what is the sieve size in millimeters?

Answer: We have included details on how soils are treated and preserved for soil chemical property analysis in the Materials and Methods section, specifically in the section on Soil Chemical Property Analysis, as per your suggestion (lines 133-139).

Comment: Lines 236-238: References need to be provided.

Answer: We have already added a reference as per your suggestion, and it can be found in line 234.

Comment: Lines 273, 326, 345: "p" may be rewritten as "P."

Answer: We have made the change to capitalize "p" as per your suggestion.

Comment: Line 285: "shows" may be rewritten as "showed."

 Answer: We have made the change to use "showed" as you recommended.

Comment: Line 425: "...[80-82]. Importantly, ..."

 Answer: We have revised as you suggested.

Sincerely,

Jiraporn Jirakkakul, Ph.D

---

## [Editor Report · Decision Letter 2]

19 Oct 2023

Soil Microbiome Analysis Reveals Effects of Periodic Waterlogging Stress on Sugarcane Growth

PONE-D-23-16598R2

Dear Dr. Jirakkakul,

We’re pleased to inform you that your manuscript has been judged scientifically suitable for publication and will be formally accepted for publication once it meets all outstanding technical requirements.

Kind regards,

Erika Kothe

Academic Editor

PLOS ONE
---

## [Editor Report · Acceptance letter]

24 Oct 2023

PONE-D-23-16598R2 

Soil Microbiome Analysis Reveals Effects of Periodic Waterlogging Stress on Sugarcane Growth 

Dear Dr. Jirakkakul:

I'm pleased to inform you that your manuscript has been deemed suitable for publication in PLOS ONE. Congratulations! Your manuscript is now with our production department. 

Kind regards, 

on behalf of

Prof. Dr. Erika Kothe 

Academic Editor

PLOS ONE